# GAN-Based Inversion of Crosshole GPR Data to Characterize Subsurface Structures

Donghao Zhang [1,2,3] , Zhengzheng Wang [1,2] , Hui Qin [1,2,3,*] , Tiesuo Geng [1,2,3] and Shengshan Pan [1,2]

1 State Key Laboratory of Coastal and Offshore Engineering, Dalian University of Technology, Dalian 116024, China; dhzhang@mail.dlut.edu.cn (D.Z.); wangzhengzheng@dlut.edu.cn (Z.W.); gengts@dlut.edu.cn (T.G.); pssbu@dlut.edu.cn (S.P.)
2 School of Civil Engineering, Dalian University of Technology, Dalian 116024, China
3 Research Institute of Dalian University of Technology in Shenzhen, Shenzhen 518057, China
* Correspondence: hqin@dlut.edu.cn

**Abstract:** The crosshole ground-penetrating radar (GPR) technique is widely used to characterize subsurface structures, yet the interpretation of crosshole GPR data involves solving non-linear and ill-posed inverse problems. In this work, we developed a generative adversarial network (GAN)-based inversion framework to translate crosshole GPR images to their corresponding 2D defect reconstruction images automatically. This approach uses fully connected layers to extract global features from crosshole GPR images and employs a series of cascaded U-Net structures to produce high-resolution defect reconstruction results. The feasibility of the proposed framework was demonstrated on a synthetic crosshole GPR dataset created with the finite-difference time-domain (FDTD) method and real-world data from a field experiment. Our inversion network obtained recognition accuracy of 91.36%, structural similarity index measure (SSIM) of 0.93, and RAscore of 91.77 on the test dataset. Furthermore, comparisons with ray-based tomography and full-waveform inversion (FWI) suggest that the proposed method provides a good balance between inversion accuracy and efficiency and has the best generalization when inverting actual measured crosshole GPR data.

**Keywords:** subsurface structure; crosshole ground-penetrating radar (GPR); inversion; deep learning; generative adversarial network (GAN); finite-difference time domain (FDTD)

## 1. Introduction

As a fast and accurate non-destructive testing method, crosshole ground-penetrating radar (GPR) has been increasingly used in deep-foundation defect detection [1], engineering geology investigation [2,3], water-resource exploration, etc. [4–6]. Crosshole GPR excites high-frequency electromagnetic waves from a transmitting antenna and collects electromagnetic waves with a receiving antenna in the boreholes of subsurface media [7]. These electromagnetic waves form refracted and reflected waves when passing through different underground media. Then, the image of the underground media can be characterized by analyzing the measured crosshole GPR data [8]. However, crosshole GPR data are not the direct imaging of a subsurface structure, and interpretation involves geophysical data inversion, which is highly complicated and time-consuming.

The commonly used ray-based tomography methods employ the first-arrival traveltimes or first-cycle amplitudes of crosshole GPR data to infer the subsurface wave velocity or attenuation distribution. These methods use just a small proportion of the information from crosshole GPR data, so they merely lead to a low-resolution inversion output, resulting in the inability to accurately characterize features with sub-wavelength [9]. On the contrary, the full-waveform inversion (FWI) method, which takes into account the entire waveform information of crosshole GPR data, is capable of providing high-resolution inversion results [10–12]. Yet, FWI requires the correct selection of the source wavelet and starting

model to prevent the inversion result from being the local minimum [9,13,14], which makes this method difficult to apply to real-world data inversion. Then, probabilistic inversion methods with Bayesian inference have been proposed [15–18]. These approaches allow for the treatment of different sources of error and return an ensemble of statistically acceptable solutions. They are capable of preventing the inversion results from being trapped in the local minimum and able to quantify the uncertainties of the inversion results. Nevertheless, the probabilistic inversion methods involve thousands to millions of forward simulations to ensure the inversion results are converged to the posterior distribution, which leads to the consumption of considerable computing resources [19,20].

Deep learning is a novel branch of machine learning [21]. With the emergence of convolutional neural networks [22,23], it has received great attention. The network used in deep learning can spontaneously learn low-order features (edges, textures, etc.) and high-order features (high-dimension features related to specific training tasks) in an image, and there is no need to manually design features in advance as required in traditional machine learning methods [24,25]. Additionally, after the appearance of the generative adversarial network (GAN) [26], the capability of deep learning is no longer limited to image classification and object detection. It can also generate virtual images that do not exist in reality from training data. By adding labels to the input of generators and discriminators or introducing a group of generators and discriminators, the improved GAN can learn the non-linear mapping relationship between different domains and realize end-to-end image conversion [27]. This capability makes it possible to translate crosshole GPR images to their corresponding dielectric properties, which are defined by permittivity or conductivity images, with deep learning models.

In recent years, some studies have applied the GAN framework to assist GPR data analysis, such as eliminating clutters in GPR images [28,29] and performing GPR-image simulation for data augmentation [30,31]. All these works are for ground-coupled GPR data, but crosshole GPR uses different antenna arrangements; therefore, the interpretation of crosshole GPR data involves more non-linear and ill-posed problems. Inspired by image-to-image translation using GANs and U-Nets [32–34], this paper proposes a GAN-based inversion network that maps crosshole GPR images to the corresponding defect reconstruction images for subsurface imaging. We first introduce the design of the inversion network and then illustrate the preparation of the dataset. Next, we investigate the impact of the hyperparameters and data formats on the inversion network in order to achieve the optimal inversion performance. Afterwards, we validate the feasibility of the proposed method with both synthetic and real-world data and compare it with tomography and FWI. Finally, we discuss the rationale of the key components and conclude this paper with the main findings.

## 2. Methodology

In this paper, we propose a GAN-based inversion network to transform crosshole GPR data into defect reconstruction images, as depicted in Figure 1. Instead of using a single GAN to invert crosshole GPR data, we adopt a two-stage strategy, namely, low- and high-resolution GANs. The low-resolution GAN extracts global features from crosshole GPR time-domain waveforms and produces a low-resolution defect reconstruction image. The high-resolution GAN enlarges the size of the above predicted map and modifies it so that the final results can be presented with high resolution.

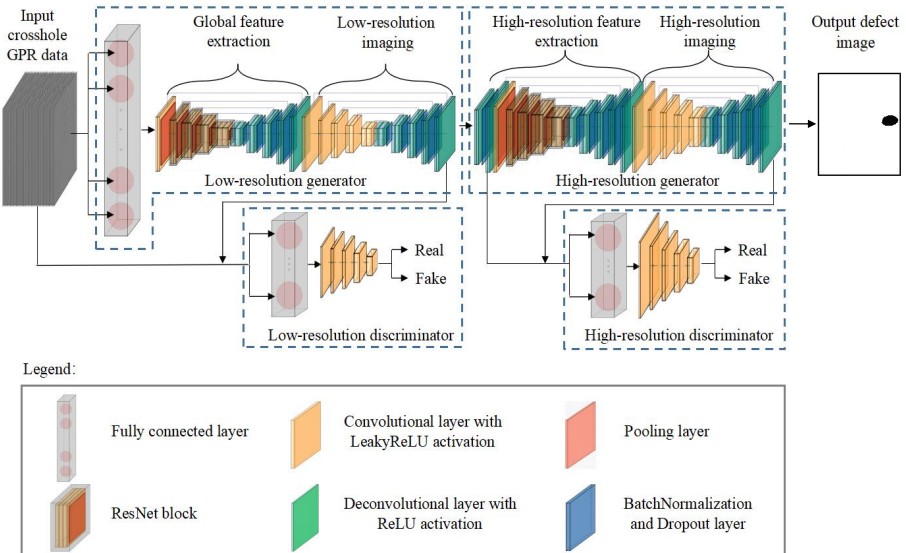

**Figure 1.** Structure of the GAN-based crosshole GPR data inversion network.

Both the low- and high-resolution GANs consist of a generator and discriminator. The generator encodes input crosshole GPR images to feature maps and then decodes the feature maps to the corresponding defect reconstruction image. The discriminator combines the input data with the generated defect reconstruction image to judge the verisimilitude of the predicted image, assigning label "1" if the result is a true defect reconstruction image or label "0" if it is a fake defect reconstruction image, and subsequently feeds the result back to the generator to update its weights. With the training process, the inversion network is optimized when the discriminator no longer differentiates the simulated defect reconstruction image from the training images [35].

The objective function of the network is defined as

$$\min_{G}\max_{D}V(D,G) = E_{x\sim P_{\text{data}}(x)}[\log(D(x))] + E_{z\sim P_z(z)}[\log(1-D(G(z)))]. \tag{1}$$

Since network training in one epoch can be divided into two steps, which are discriminator training and generator training, Equation (1) can be disassembled into the following two equations:

$$\max_{D}V(D,G) = E_{x\sim P_{\text{data}}(x)}[\log(D(x))] + E_{z\sim P_z(z)}[\log(1-D(G(z)))] \tag{2}$$

and

$$\min_{G}V(D,G) = E_{z\sim P_z(z)}[\log(1-D(G(z)))], \tag{3}$$

where $\min_{G}V(D,G)$ is the discriminator's performance value function, representing the discriminator's ability to distinguish between true and false images. The smaller the function value is, the better the discriminator performs. $\max_{D}V(D,G)$ denotes the generator's performance value function, which indicates the difference between the virtual image generated by the generator and the real image. The greater the function value is, the better the generator performs. $x$ is a real sample that obeys the distribution of the real dataset. $z$ is an input sample of the generator that follows the distribution of the generator's input data. $E_{x\sim P_{\text{data}}(x)}[\log(D(x))]$ is a mathematical expectation constructed according to the logarithmic loss of the discriminator identifying the real samples. When the performance of the discriminator is optimized, the discriminator gives the judgment of "1" for all images in the real dataset. $E_{z\sim P_z(z)}[\log(1-D(G(z)))]$ is the logarithmic loss of virtual samples made based on the discriminator's judgment. Similarly, in the optimal state, the discriminator labels all images in the virtual dataset with "0", that is, $D(G(z)) = 0$.

In summary, the network first optimizes the performance of the discriminator to maximize the mathematical expectation of the two logarithmic losses in Equation (1). Then, it optimizes the performance of the generator by producing virtual samples to obtain real labels from the discriminator, that is, $D(G(z)) = 1$. Therefore, the optimization process of the generator's performance is to minimize the mathematical expectation of the logarithmic loss in Equation (3). It can be seen from Equations (2) and (3) that network training is a game between discriminator and generator. The objective function obtains its optimal value when the discriminator cannot distinguish the fake images produced by generator from real images. In this state, the discriminator gives all images a score of 0.5.

## 2.1. Low-Resolution GAN

In the low-resolution GAN, the generator consists of a fully connected (FC) layer followed by two U-Nets in series [34]. Among them, the first U-Net and fully connected layer are called the global feature extraction network, which is responsible for extracting and encoding global features of crosshole GPR data. Afterwards, the features are fed to the second U-Net to be decoded into a low-resolution defect reconstruction image of enlarged size.

A crosshole GPR image is first input into a fully connected layer composed of 1024 neuron nodes, in which each neuron is connected with all pixels of the input data with a weight assigned to each connection. Thus, the fully connected layer processes the global information of the input data, combines the outputs of the 1024 neurons, and generates a complex high-dimension global feature map of the input images with sizes $64 \times 64 \times 1024$.

Then, we use a U-Net structure named global feature extraction network to merge the features obtained from the fully connected layer and reduce the dimensionality of the feature map. The global feature extraction network employs the first 100 layers of ResNet101 V2 as the encoder and uses 12 cascade deconvolutional layers (Deconv2d) as the decoder. Once processed by the global feature extraction network, the dimensionality of the feature map is reduced to $64 \times 64 \times 32$.

Afterwards, a low-resolution inversion network is used to further filter redundant features and infer the defect reconstruction image based on the features created by the global feature extraction network. The decoder and encoder of the low-resolution inversion network are symmetrically arranged with 10 cascaded convolutional (Conv2d) and deconvolutional layers, respectively. Meanwhile, in order to suppress over-fitting and speed up the convergence rate during network training, we placed a batch normalization (BN) layer and a dropout layer between every two deconvolutional layers. In addition, we set horizontal connections to link same-size features between the convolutional and deconvolutional layers. This operation utilizes low- and high-order feature maps comprehensively to suppress possible gradient vanishing. Once having passed through the low-resolution inversion network, the feature maps are converted into a low-resolution defect reconstruction image with the size of $64 \times 64 \times 1$.

Finally, we utilize a neural network with a fully connected layer of 256 neurons and five cascaded convolutional layers as the low-resolution discriminator, which receives both crosshole GPR data and the low-resolution defect image and then gives a $4 \times 4$ score matrix to the low-resolution defect image.

## 2.2. High-Resolution GAN

The low-resolution GAN establishes an end-to-end non-linear relationship between crosshole GPR data and the corresponding low-resolution defect reconstruction image that has only $64 \times 64$ pixels. In order to image the inverted targets more precisely, we designed an additional high-resolution GAN to refine the defect reconstruction image derived from the low-resolution GAN and improve the inversion resolution to $256 \times 256$.

The task of the high-resolution GAN is no longer to establish the mapping relationship between the crosshole GPR image and the defect reconstruction image. Instead, it is used to denoise and enhance the resolution of the low-resolution defect reconstruction

image. In the high-resolution GAN, a U-Net structure named high-resolution feature extraction network is used to extract high-resolution features from the defect reconstruction images produced by the low-resolution GAN. The image is first enlarged to $256 \times 256 \times 1$ by four deconvolutional layers and then fed into ResNet50 V2 for high-order feature extraction. Then, 12 cascaded deconvolutional layers are implemented to enlarge the size of the extracted feature map to $256 \times 256 \times 1$. Horizontal connections are also set between the same-size feature maps of ResNet50 V2 and the deconvolutional network to comprehensively utilize the feature vectors in different stages and suppress gradient vanishing. After high-resolution feature extraction, we use a high-resolution inversion network with the same structure as the low-resolution inversion network to invert the high-resolution features to high-resolution defect reconstruction images.

We also use a neural network with a fully connected layer of 256 neurons and five cascaded convolutional layers as the high-resolution discriminator to form the GAN architecture. The high-resolution discriminator is employed to judge the authenticity of the high-resolution defect images.

Table 1 summarizes the main layers in the proposed framework and the shape of each layer. For each training batch, the shape of the feature map flowing in the network is three-dimensional, and the three dimensions are width, height, and channel (W, H, C), respectively. In general, the larger the number of channels in the feature map, the more abstract the information contained in the feature map [25].

**Table 1.** Architecture of the proposed network.

| Low-Resolution Generator | | High-Resolution Generator | |
|---|---|---|---|
| Model Layer | Output Shape (W, H, C) | Model Layer | Output Shape (W, H, C) |
| FC_1 | (64, 64, 1024) | Deconv2d_16 | (128, 128, 8) |
| ResNet 101 | (2, 2, 2048) | Deconv2d_17 | (256, 256, 16) |
| Deconv2d_1 | (4, 4, 256) | ResNet 50 | (8, 8, 2048) |
| BN_1 and Dropout_1 | (4, 4, 256) | Deconv2d_18 | (8, 8, 256) |
| Deconv2d_2 | (4, 4, 256) | BN_14 and Dropout_14 | (8, 8, 256) |
| BN_2 and Dropout_2 | (4, 4, 256) | Deconv2d_19 | (16, 16, 128) |
| Deconv2d_3 | (8, 8, 64) | BN_15 and Dropout_15 | (16, 16, 128) |
| BN_3 and Dropout_3 | (8, 8, 64) | Deconv2d_20 | (16, 16, 128) |
| Deconv2d_4 | (8, 8, 64) | BN_16 and Dropout_16 | (16, 16, 128) |
| BN_4 and Dropout_4 | (8, 8, 64) | Deconv2d_21 | (32, 32, 64) |
| Deconv2d_5 | (16, 16, 64) | BN_17 and Dropout_17 | (32, 32, 64) |
| BN_5 and Dropout_5 | (16, 16, 64) | Deconv2d_22 | (32, 32, 64) |
| Deconv2d_6 | (16, 16, 64) | BN_18 and Dropout_18 | (32, 32, 64) |
| BN_6 and Dropout_6 | (16, 16, 64) | Deconv2d_23 | (64, 64, 32) |
| Deconv2d_7 | (32, 32, 64) | BN_19 and Dropout_19 | (64, 64, 32) |
| BN_7 and Dropout_7 | (32, 32, 64) | Deconv2d_24 | (64, 64, 32) |
| Deconv2d_8 | (32, 32, 64) | BN_20 and Dropout_20 | (64, 64, 32) |
| BN_8 and Dropout_8 | (32, 32, 64) | Deconv2d_25 | (128, 128, 16) |
| Deconv2d_9 | (64, 64, 32) | BN_21 and Dropout_21 | (128, 128, 16) |
| BN_9 and Dropout_9 | (64, 64, 32) | Deconv2d_26 | (128, 128, 16) |
| Deconv2d_10 | (64, 64, 32) | BN_22 and Dropout_22 | (128, 128, 16) |
| Conv2d_1 | (32, 32, 16) | Deconv2d_26 | (256, 256, 8) |
| Conv2d_2 | (16, 16, 32) | BN_23 and Dropout_23 | (256, 256, 8) |
| Conv2d_3 | (8, 8, 64) | Deconv2d_26 | (256, 256, 1) |
| Conv2d_4 | (4, 4, 128) | Conv2d_6 | (128, 128, 16) |
| Conv2d_5 | (4, 4, 256) | Conv2d_7 | (64, 64, 32) |
| Deconv2d_11 | (4, 4, 256) | Conv2d_8 | (32, 32, 64) |
| BN_10 and Dropout_10 | (4, 4, 256) | Conv2d_9 | (16, 16, 128) |
| Deconv2d_12 | (8, 8, 128) | Conv2d_10 | (16, 16, 256) |
| BN_11 and Dropout_11 | (8, 8, 128) | Deconv2d_27 | (16, 16, 256) |
| Deconv2d_13 | (16, 16, 64) | BN_24 and Dropout_24 | (16, 16, 256) |
| BN_12 and Dropout_12 | (16, 16, 64) | Deconv2d_28 | (16, 16, 128) |

**Table 1.** *Cont.*

| Low-Resolution Generator | | High-Resolution Generator | |
| --- | --- | --- | --- |
| **Model Layer** | **Output Shape (W, H, C)** | **Model Layer** | **Output Shape (W, H, C)** |
| Deconv2d_14 | (32, 32, 32) | BN_25 and Dropout_25 | (16, 16, 128) |
| BN_13 and Dropout_13 | (32, 32, 32) | Deconv2d_29 | (32, 32, 64) |
| Deconv2d_15 | (64, 64, 1) | BN_26 and Dropout_26 | (32, 32, 64) |
| - | - | Deconv2d_30 | (64, 64, 32) |
| - | - | BN_27 and Dropout_27 | (64, 64, 32) |
| - | - | Deconv2d_31 | (128, 128, 16) |
| - | - | BN_28 and Dropout_28 | (128, 128, 16) |
| - | - | Deconv2d_32 | (256, 256, 1) |
| - | - | BN_29 and Dropout_29 | (256, 256, 1) |
| **Low-Resolution Discriminator** | | **High-Resolution Discriminator** | |
| **Model Layer** | **Output Shape (W, H, C)** | **Model Layer** | **Output Shape (W, H, C)** |
| FC_2 | (64, 64, 256) | FC_3 | (256, 256, 256) |
| Conv2d_11 | (32, 32, 64) | Conv2d_16 | (128, 128, 64) |
| Conv2d_12 | (16, 16, 128) | Conv2d_17 | (64, 64, 128) |
| Conv2d_13 | (8, 8, 256) | Conv2d_18 | (32, 32, 256) |
| Conv2d_14 | (4, 4, 512) | Conv2d_19 | (16, 16, 512) |
| Conv2d_15 | (4, 4, 1) | Conv2d_20 | (16, 16, 1) |

In addition, we use the Gaussian blur algorithm to remove random noise from the inverted defect reconstruction images and the Canny edge detection algorithm to extract pixel coordinates of the target edge and obtain the target centroid position [36]. Figure 2a presents an example of a GAN-inverted defect reconstruction image that contains random noise. After having been processed by the Gaussian blur algorithm, the noise was suppressed, yet the target edge became unclear (Figure 2b). Then, the Canny edge detection algorithm was used to extract the target boundary (Figure 2c), so that the position and shape of the target could be determined. In this work, a 15 × 15 Gaussian kernel with zero standard deviation was used to remove random noise, and the weak and strong edge thresholds of the Canny edge detector were 50 and 150, respectively.

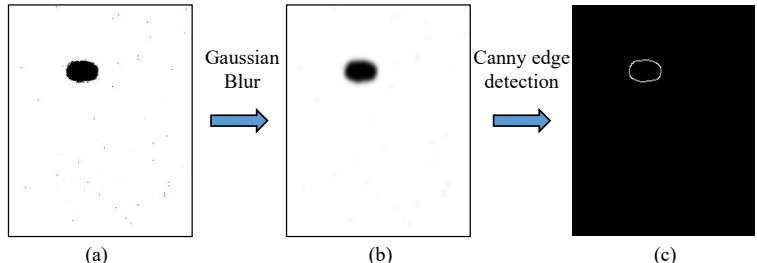

**Figure 2.** Target edge detection process: (**a**) inverted defect reconstruction image, (**b**) Gaussian-blur-algorithm-processed image, and (**c**) Canny-edge-detection algorithm-processed image.

### 2.3. Network Performance Evaluation

We use the mean square error (MSE) as the loss function to evaluate the difference between the predicted defect reconstruction image and the ground truth; it is defined as

$$\mathrm{MSE}(P_{\mathrm{pre}}, P_{\mathrm{real}}) = \frac{1}{m}\sum_{i=1}^{m}(p_{\mathrm{pre},i} - p_{\mathrm{real},i})^2, \tag{4}$$

where $P_{\mathrm{pre}}$ and $P_{\mathrm{real}}$ denote the predicted and real images, respectively, and $p_{\mathrm{pre},i}$ and $p_{\mathrm{real},i}$ are the $i$-th pixel values of the predicted and real images, respectively. In the training stage,

the MSE is calculated in each step to guide the weight parameters to be updated. The network reaches the optimum when the MSE no longer decreases [37].

We also introduce the structural similarity index measure (SSIM) to comprehensively evaluate the reconstruction ability of the inversion network [38,39]; it is defined as

$$\text{SSIM}(P_{\text{pre}}, P_{\text{real}}) = \left(\frac{2\mu_{\text{pre}}\mu_{\text{real}} + C_1}{\mu_{\text{pre}}^2 + \mu_{\text{real}}^2 + C_1}\right)^\alpha + \left(\frac{2\sigma_{\text{pre}}\sigma_{\text{real}} + C_2}{\sigma_{\text{pre}}^2 + \sigma_{\text{real}}^2 + C_2}\right)^\beta + \left(\frac{2\sigma_{\text{pre,real}} + C_3}{\sigma_{\text{pre}}\sigma_{\text{real}} + C_3}\right)^\gamma, \quad (5)$$

where $\mu_{\text{pre}}$ and $\mu_{\text{real}}$ are the mean pixel values of the predicted and real images, respectively; $\sigma_{\text{pre}}$ and $\sigma_{\text{real}}$ represent the pixel variances of the predicted and real images, respectively; $\sigma_{\text{pre,real}}$ is the covariance between the predicted and real images; $C_1$, $C_2$, and $C_3$ are constant coefficients that prevent the denominators from reaching zero; and $\alpha$, $\beta$, $\gamma$ are coefficients, with $\alpha\beta\gamma = 1$. The expressions $\left(\frac{2\mu_{\text{pre}}\mu_{\text{real}} + C_1}{\mu_{\text{pre}}^2 + \mu_{\text{real}}^2 + C_1}\right)^\alpha$, $\left(\frac{2\sigma_{\text{pre}}\sigma_{\text{real}} + C_2}{\sigma_{\text{pre}}^2 + \sigma_{\text{real}}^2 + C_2}\right)^\beta$, and $\left(\frac{2\sigma_{\text{pre,real}} + C_3}{\sigma_{\text{pre}}\sigma_{\text{real}} + C_3}\right)^\gamma$ quantify the brightness, contrast, and structural similarities between the predicted and real images, respectively.

When evaluating the performance of the inversion network, the inversion quality is given by the following equations [30]:

$$\text{RAscore} = \varphi \times \text{ACC} + (1 - \varphi) \times \text{SSIM} \qquad (6)$$

$$\text{ACC} = \frac{\text{TP}}{\text{TP} + \text{FP} + \text{FN}}, \qquad (7)$$

where RAscore measures the recognition ability of the inversion network and ACC computes the recognition accuracy on the test dataset. TP refers to the true-positive samples representing the recognized samples with correct target locations; FP denotes the false-positive samples with wrong target positions; and FN refers to the false-negative samples that fail to identify the targets. $\varphi$ is a weighting coefficient used to balance the weights of ACC and SSIM in Equation (6), which was set to 0.75 in this work.

We consider that the maximum error limit for defect position recognition should be less than 5% of the side length of the defect reconstruction image, which is 12.8 pixels for a 256 × 256 image. Therefore, we used 10 pixels as the threshold value in this work. If the distance between the centroid position of the inverted target and the true value is within the threshold value, the inversion result is regarded as a TP. Otherwise, an FP label is assigned. In addition, if no corresponding target information is inverted within 10 pixels around the actual target position, the inverted image is classified as an FN.

It needs to be noted that direct inversion with high resolution introduces a huge number of weights in the fully connected layer, resulting in graphics memory explosion. Thus, we first use the low-resolution GAN to invert low-resolution crosshole GPR images (64 × 64) to the same-size defect reconstruction image with noise and false-positive samples, then use the high-resolution GAN to enlarge the size of the defect reconstruction image to 256 × 256, and refine it by removing noise and small false-positive samples. Figure 3a depicts the training losses of the low- and high-resolution GANs, in which the MSE value of the low-resolution GAN reaches approximately 1400, while the high-resolution GAN further improves the MSE value to around 400. This indicates that the high-resolution GAN greatly optimizes the low-resolution results. Figure 3b demonstrates the inverted defect reconstruction images of the low- and high-resolution GANs, respectively. In the first step, the low-resolution GAN translates the crosshole GPR images to a low-resolution defect reconstruction image with considerable noise. In the second step, the high-resolution GAN enhances the resolution of the defect reconstruction image and reduces the noise tremendously.

Low-resolution GAN

High-resolution GAN

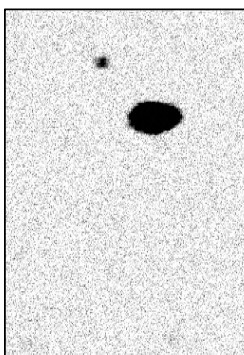 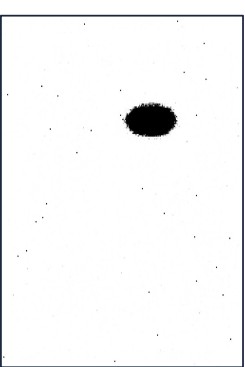

**Figure 3.** Evaluation of the inversion effects of the low- and high-resolution GANs.

## 3. Dataset

The inversion network requires a large number of labeled data to train the model parameters, yet it is difficult to collect sufficient qualified crosshole GPR data in real-world measurements. On one hand, due to lack of knowledge of the subsurface structure, the collected data cannot be guaranteed to be consistent with the corresponding permittivity distribution. On the other hand, measured data always have noise that may adversely affect network training. In consideration of the above factors, we used the finite-difference time-domain (FDTD) method to create simulated crosshole GPR data from known permittivity distributions and prepared crosshole GPR waveform and defect distribution data pairs to form the training and test datasets for the inversion network.

As a demonstration, we took underground diaphragm wall defect detection with crosshole GPR as an example to prepare the datasets. Referring to the structural properties of diaphragm walls in real-world applications, we constructed 1100 3D numerical diaphragm wall models with dimensions of 2.00 m × 4.00 m × 0.40 m using gprMax software (v3.1.5, Dr Craig Warren, Edinburgh, UK) [40], as shown in Figure 4. Each model is spatially discretized with cubic grids of 0.005 m in edge length. The perfectly matched layer (PML) was adopted as the boundary condition, with 10 grid cells in each dimension. The material of the diaphragm walls was defined as concrete with relative permittivity of 9 and electric conductivity of 0.001 S/m. Inside the diaphragm walls, we put defects of different sizes and shapes with relative permittivity of 20 and electric conductivity of 0.01 S/m as targets. As both the concrete walls and defects were non-magnetic materials, the relative permeability and magnetic loss were set to 1 and 0 $\Omega$/m, respectively.

The 1100 forward models simulated five different defect situations, which were a single cylinder defect, a single cuboid defect, two cylinder defects, two cuboid defects, and one cylinder defect and one cuboid defect. Each situation contained 220 numerical models. The diameters or side lengths of the defects were randomly generated in the range from one-quarter to one time the crosshole GPR wavelength (a center frequency of 500 MHz was used in the simulations), and the center positions were distributed randomly in the diaphragm wall models.

For each diaphragm wall model, 36 transmitting- and 36 receiving-antenna measurement points were placed on the left and right sides of the model to collect crosshole GPR data. The wave source of the antenna was the Ricker wavelet, with a center frequency of 500 MHz. The detection depth range was 0.2 m to 3.8 m, while the distance between the measuring lines of the transmitting and receiving antennas was 1.8 m. In one crosshole GPR measurement, the transmitting antenna emitted electromagnetic waves in one borehole, and the receiving antenna collected data in the adjacent borehole with a sampling interval of 0.1 m. We sampled the electromagnetic waves on each discretized grid, and the number of the A-scan sampling points was 5194, with a time window of 50 ns. In this way, a time-domain crosshole GPR image was formed, as illustrated in Figure 5a. After repeating the same sampling process for the transmitting-antenna measurement points at

36 depths, we obtained 36 crosshole GPR B-scan images. The detailed modeling parameters are summarized in Table 2.

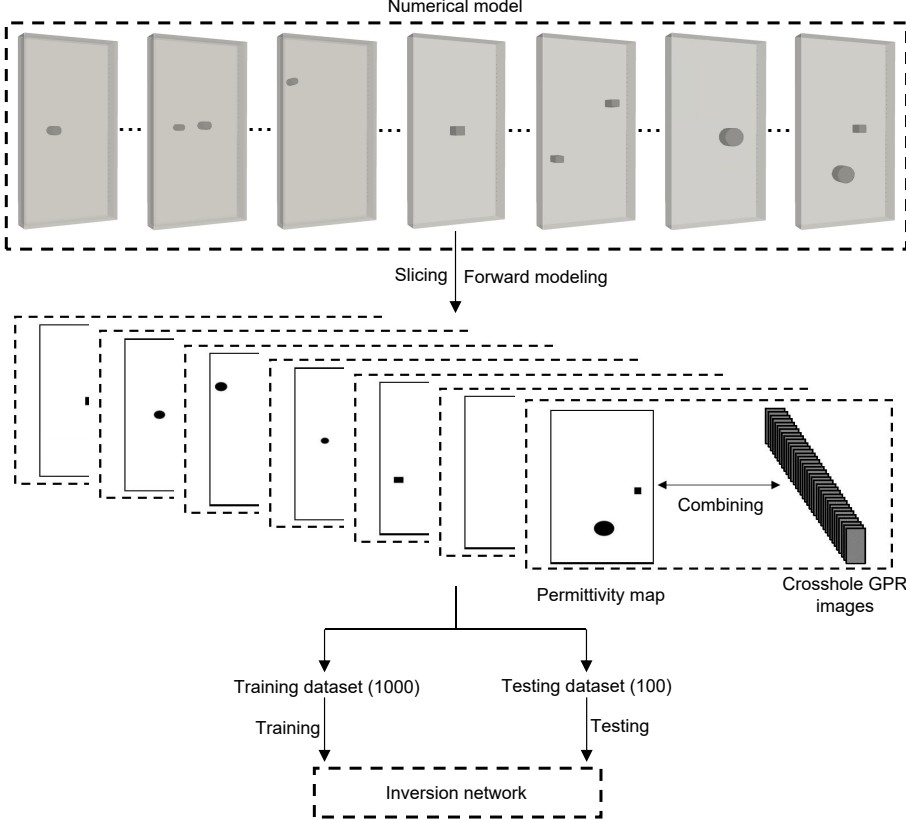

**Figure 4.** Crosshole GPR dataset preparation using numerical simulations.

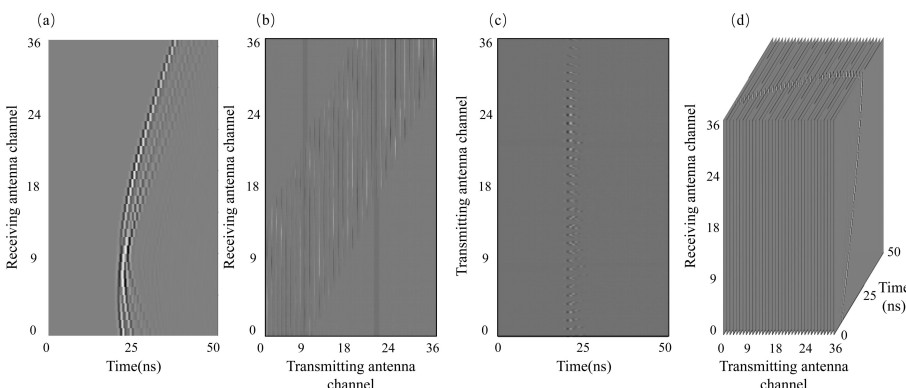

**Figure 5.** Different crosshole GPR data combination modes: (**a**) a single image from the measurement of one emitter position and multiple receiver positions, (**b**–**d**) images assembled in the width, height, and channel directions, respectively.

In the next step, the 36 crosshole GPR images needed to be assembled into a single image to form the training dataset. Possibly, these images could be combined in the width, height, or channel direction, as shown in Figure 5b–d. In the field of medical imaging, data are usually superimposed in the width direction [41], but this suppresses features in the traveltime dimension. To stack images in the height direction is customary in the field of geophysics, but this reduces the number of data in the depth direction. In this paper, we chose another option, i.e., to superimpose the crosshole GPR images in the channel direction from a 3D matrix, which preserves all the features without compression.

**Table 2.** Summary of the FDTD modeling parameters.

| Modeling Parameter | Value |
|---|---|
| Model size (width × depth × thickness) | 2.00 m × 4.00 m × 0.40 m |
| Discretized grid | 0.005 m × 0.005 m × 0.005 m |
| Antenna detection depth range | 0.2–3.8 m |
| Transmitting–receiving baseline distance | 1.8 m |
| Sampling interval | 0.1 m |
| Source wavelet | Ricker |
| Center frequency | 500 MHz |
| Time window | 50 ns |
| Sampling points for each A-scan | 5194 |

In this work, as the dimensions of defects along the thickness direction were fixed, we sliced a 2D permittivity profile along the length direction of each numerical model to represent the permittivity distribution of the 3D model. Meanwhile, in order to reduce redundant data and speed up the training speed of the deep learning model, we transformed the permittivity images to binary pixel images, and we assigned "1" (white) to the pixels of concrete and gave "0" (black) to the pixels of defects. As shown in Figure 4, the crosshole GPR images and the corresponding 2D binary defect reconstruction images were assembled to form the training and test datasets. Then, we added Gaussian noise to these simulated images and constructed data pairs with their corresponding defect reconstruction images for network training and testing. Each data pair incorporated 36 crosshole GPR images and one corresponding defect reconstruction image. In this way, the relationship between crosshole GPR data and the defect reconstruction image was established.

## 4. Network Training and Testing

We trained the GAN-based framework on the training dataset for 500 epochs. We used the Adam optimizer with a learning rate of 0.01 and a normal distribution initializer to start the training process. It took 19 h and 37 min to train the network for 500 epochs on a desktop PC equipped with Intel i9-9920X CPU (sourced from Intel Corporation, Santa Clara, CA, USA) and NVIDIA RTX 2080 Ti graphics card (sourced from NVIDIA Corporation, Santa Clara, CA, USA). Figure 6 shows the training losses of the low- and high-resolution GANs. The loss curves of the generators decreased sharply in the first 80 epochs and converged gradually to around 200 at the 500th epoch, indicating that the image quality of both the low- and high-resolution generators gradually improves with the training epochs, as shown in Figure 6a. Moreover, the MSE value of the low-resolution generator reached approximately 1400, while the high-resolution GAN further reduced the MSE value to around 400. This demonstrates that the high-resolution GAN greatly optimizes the low-resolution results. Figure 6b depicts the loss curves of the low- and high-resolution discriminators when distinguishing real and fake images. The loss values of the discriminators gradually stabilized with the training epochs, and all converged to around 0.25 at the 500th epoch. This was mainly because the two discriminators failed to tell apart the generated fake images and the real images. As a result, they gave a neutral score of 0.5 to both the real and fake images, which corresponds to an MSE value of 0.25. In general, we can infer from the training curves that the generators and the discriminators can reach a balanced state at the 500th epoch, which demonstrates that the GAN-based framework achieves the ideal training effect.

After training, we evaluated network performance on the test dataset, which contained 100 numerical models. We computed the ACC, SSIM, and RAscore of the network, and the test metrics were consistent with the findings of the training process. On the test dataset, we obtained the network performance of ACC = 91.36%, SSIM = 0.93, and RAscore = 91.77.

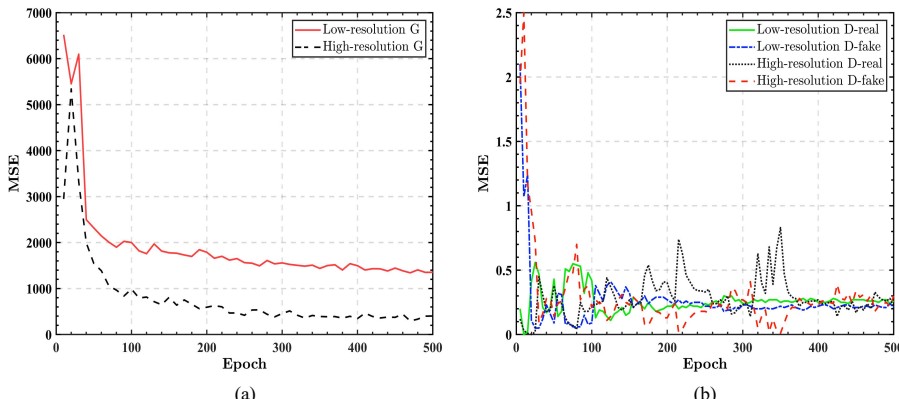

**Figure 6.** Training losses of the proposed framework: (**a**) loss curves of generators (G) and (**b**) loss curves of discriminators (D).

## 5. Inversion Results

### 5.1. Synthetic Data Inversion Results

As shown in Figure 7, we created seven test numerical models with dimensions of 2.00 m × 4.00 m × 0.40 m to simulate diaphragm walls with relative permittivity of 9 and electric conductivity of 0.001 S/m. In cases 1 to 4, a cuboid defect with relative permittivity of 20 and electric conductivity of 0.01 S/m was embedded inside each model to simulate the target. In order to evaluate the inversion resolution of the network, we set the sizes of the targets to one time, half, and one-quarter the wavelength, respectively, and considered a target of one-quarter the wavelength in the corner of the model with sparse data coverage. We also considered multiple-target inversion. In the last three cases, two defects were arranged horizontally, vertically, and diagonally in the numerical models, respectively. We simulated the crosshole GPR measurements using a 500 MHz source wavelet, and data were collected in the same way as in network training.

Then, we utilized these cases to compare our inversion network with two classic crosshole GPR data inversion methods, including ray-based tomography and FWI [10,42]. From the second to the last rows in Figure 7 display the ground truth and the inverted defect reconstruction images of different methods, respectively. FWI demonstrated the best inversion capability, as it achieved the highest precision in all cases in reconstructing the targets. On the other hand, ray-based tomography failed to resolve targets smaller than half the wavelength and was plagued by considerable noise. Our method used fully connected layers and horizontal connections to extract global features and correctly pinpointed almost all target locations.

Table 3 summarizes MSE, SSIM, and time cost of the inversion results to quantitatively evaluate the inversion performance of different methods. It is evident that FWI had the best MSE and SSIM performance in all the cases compared with the other methods, yet it suffered from considerable longer inversion time. Although the MSE values of the proposed method were apparently greater than those of FWI, the SSIM values were slightly lower than those of FWI (~7%), indicating that the reconstructing capability of the proposed method was close to that of FWI. Notably, the inversion time of the proposed method was significantly shorter than that of FWI, which is crucial in large-scale inversion tasks. Compared with ray-based tomography, both the MSE and SSIM performance results of the proposed method were much better, with similar time costs.

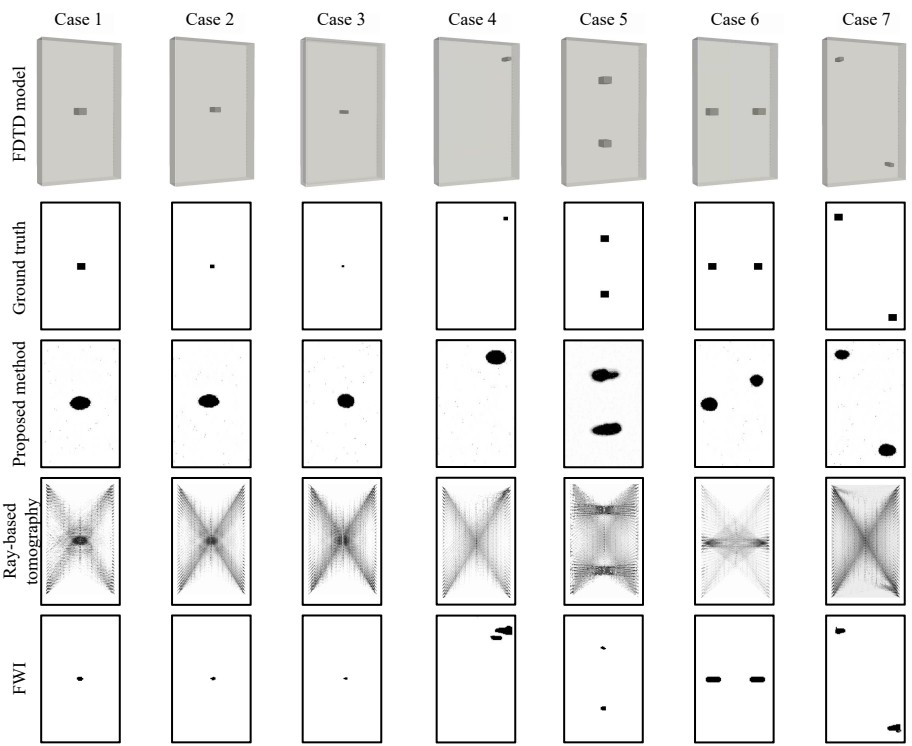

**Figure 7.** Inversion results of synthetic crosshole GPR data.

**Table 3.** Performance indicators of different inversion and image segmentation methods.

| Method | Indicator | Case Number | | | | | | |
|---|---|---|---|---|---|---|---|---|
| | | 1 | 2 | 3 | 4 | 5 | 6 | 7 |
| Proposed method | MSE | 1024.29 | 1273.77 | 1167.94 | 1465.12 | 2543.65 | 1918.53 | 1640.71 |
| | SSIM | 0.93 | 0.93 | 0.93 | 0.92 | 0.85 | 0.91 | 0.91 |
| | Time (s) | 2.4 | 2.3 | 2.5 | 2.5 | 2.7 | 2.5 | 2.5 |
| Ray-based tomography | MSE | 2625.57 | 3063.00 | 4130.67 | 2178.10 | 4967.70 | 1440.1 | 5136.34 |
| | SSIM | 0.34 | 0.33 | 0.30 | 0.30 | 0.25 | 0.43 | 0.23 |
| | Time (ns) | 4.5 | 4.6 | 4.6 | 4.7 | 4.6 | 4.8 | 4.7 |
| FWI | MSE | 231.76 | 81.4 | 53.67 | 871.19 | 634.41 | 506.46 | 713.47 |
| | SSIM | 0.99 | 0.99 | 0.99 | 0.97 | 0.98 | 0.98 | 0.97 |
| | Time (s) | 29.3 | 29.5 | 29.5 | 29.6 | 30.0 | 29.9 | 29.9 |

*5.2. Experiment Data Inversion Results*

A field experiment was carried out to collect real-world crosshole GPR data. The experiment site is shown in Figure 8a: a 5 m long, 3 m wide, and 5 m deep foundation pit enclosed with four 0.35 m thick concrete retaining walls. Inside one panel of the retaining walls, a cube-shaped defect with side length of 0.15 m was arranged, as depicted in Figure 8b. We performed crosshole GPR measurements through two boreholes placed in the retaining wall with spacing of 1 m. The transmitting antenna emitted electromagnetic waves in one borehole within the depth range of 1.0 to 3.9 m with step size of 0.1 m, while the receiving antenna received signals in the adjacent borehole within the depth range of 1.0 to 3.8 m with the same step size. As a result, we obtained a total of 870 A-scan traces, which formed 30 B-scan images, as depicted in Figure 8c.

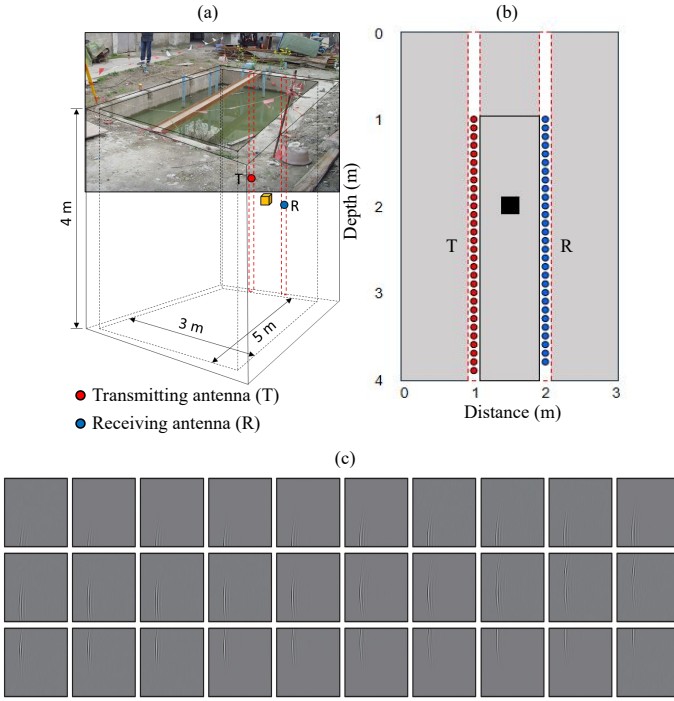

**Figure 8.** Crosshole GPR measurement: (**a**) collection scene of diagram wall, (**b**) detection model and measuring points, (**c**) measured crosshole GPR data.

Then, we input the measured crosshole GPR data into the trained GAN-based network, ray-based tomography, and FWI, respectively. The output defect reconstruction images of different methods are shown in Figure 9. The inversion network successfully revealed the location of the defect with the position error of 4.97 cm. Compared with ray-based tomography and FWI, the proposed method could generate clearer results when inverting real crosshole GPR data. Quantitatively, our method had the lowest MSE (981.19) and the highest SSIM (0.92), which were superior to those of ray-based tomography (MSE: 5547.35; SSIM: 0.80) and FWI (MSE: 5727.27; SSIM: 0.84). Since FWI requires the correct selection of initial model and source wavelet, which is easy to implement in a numerical simulation dataset but difficult to achieve in a measured dataset, although FWI had the best performance in the inversion of simulated data, it performed worse than the proposed method when inverting real-world crosshole GPR data. Note that this network is trained on a synthetic dataset, demonstrating that our inversion framework has a good generalization ability to invert real-world data.

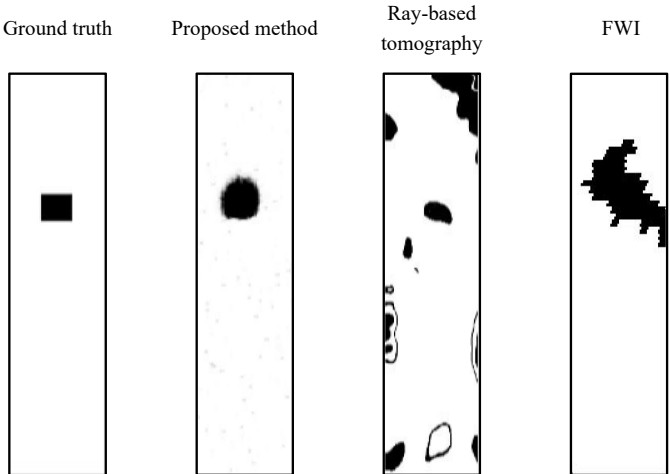

**Figure 9.** Results of inversion of real crosshole GPR data.

## 6. Conclusions

In this work, we introduce a GAN-based end-to-end inversion framework to translate crosshole GPR images to the corresponding defect reconstruction images automatically. This method uses, as building blocks, a low-resolution GAN, to extract global features from crosshole GPR images and infer a preliminary defect reconstruction image with low resolution, and a high-resolution GAN, to refine and improve the resolution of the inverted defect reconstruction image. We built an FDTD dataset based on the defect detection of diaphragm walls to train and test the proposed method. The GAN-based framework can be trained to the ideal state in which the discriminators fail to distinguish the fake images output by the generators from the real images in the training dataset. By evaluating the inversion performance of the network on a test dataset, the accuracy, SSIM, and RAscore of the optimized inversion network reached 91.36%, 0.93, and 91.77, respectively. Applications in synthetic and real-world crosshole GPR data inversion also demonstrate the applicability of the proposed method.

Our work makes contributions to the field of crosshole GPR data inversion by introducing a novel end-to-end inversion framework based on GANs. Our framework can handle complex scenarios, such as non-linearity and ill-posedness in crosshole GPR data inversion, and produce a fast, accurate, and robust method that can automatically translate crosshole GPR images to defect reconstruction images without requiring any prior knowledge or human intervention. Our framework also has the potential to be extended to other geophysical inversion problems.

We also note that when there are multiple targets, the inversion accuracy of our network declines evidently, especially when the targets are placed horizontally or vertically. In our future research, we will improve our method to accommodate multiple-target situations. Additionally, different target categories (targets with different dielectric properties) will be considered more comprehensively in our subsequent work.

**Author Contributions:** Conceptualization, methodology, writing—original draft preparation, D.Z.; data curation, software, Z.W.; writing (review and editing), funding acquisition, H.Q.; formal analysis, validation, T.G.; supervision, S.P. All authors have read and agreed to the published version of the manuscript.

**Funding:** This research was funded by National Natural Science Foundation of China (grant No. 41904095), Special Funds for Central Government Guidance to Local Governments for Science and Technology Development in Shenzhen (grant No. 2021Szvup020), Central Guidance on Local Science and Technology Development Fund of Liaoning Province (grant No. 2023JH6/100100054), and Guided Independent Research Fund of State Key Laboratory of Coastal and Offshore Engineering (grant No. SL2203).

**Data Availability Statement:** The source code and data used in this research can be publicly accessed at https://github.com/ZhangDonghao1907/GAN-based-inversion-network-for-crosshole-GPR (accessed on 2 May 2023).

**Conflicts of Interest:** The authors declare that they have no known competing financial interests or personal relationships that could have appeared to influence the work reported in this paper.

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
