# Peer review of "GAN-Based Inversion of Crosshole GPR Data to Characterize Subsurface Structures"

_remotesensing, doi:10.3390/rs15143650_

Round 1

Reviewer 1 Report

This is a relevant and readable paper. I have just a few comments and clarifications to improve it.

Line 62-70: There are recent papers also describing GANs for GPR. Please include at least a couple to show how this work is different.

Figure 1 and 2: Are two figures necessary? Figure 2 is more informative.

Line 111: "Balanced state" should be more precisely defined

No description is given of the discriminator structure.

Line 196-201: This step should be described earlier when discussing the model structure.

Section 3: What is the type of GPR pulse? Sampling rate? Number of samples? Readers would like a clear sense of the input datacube dimensions.

Section 4.5: The data structure should be mentioned when discussing the GPR in section 3. 

Reviewer 2 Report

This manuscript proposed to translate the data of cross-hole GPR to the corresponding permittivity map automatically by GAN. The experiments in the article present the influence of different factors on networks in detail. But maybe several cases are not necessary. The text is so tedious that readability needs to be improved. The manuscript cannot be accepted without major modification.

1.     First of all, authors mentioned ‘we developed a generative adversarial network (GAN) based inversion framework to translate the crosshole GPR images to their corresponding 2D permittivity map automatically’. This statement is not accurate. Because during the data mapping, the author binarizes the model, thereby eliminating the influence of permittivity.

2.     The content of the introduction still needs to be strengthened. Can describe the current development of GAN and whether there are discussions on hyper parameters in GAN.

3.     In addition, why did the author choose GAN to improve image resolution instead of the already mature VDSR?

4.     Figure 1 may miss some layers. The article mentions the use of Dropout layers, activation layers, etc. Representing it in the diagram is more conducive to readers' reproduction and learning.

5.     The representation of data types should be more standardized. For example, what does each dimension of a feature map represent?

6.     The model construction in the dataset should be described in more detail, such as the depth range and size range of the model.

7.     In section 4.1, authors compared different optimizers, but other parameters did not be given. All the parameters in each experiment should be supplied.

8.     In my opinion, some comparations are not necessary and do not have representation. Maybe authors can streamline the content of the article.

9.     The author did not provide a loss curve for the GAN network. Please supplement. The training time of the same network should also be supplemented.

10.   How does the author explain the phenomenon of MSE curve increasing first and then decreasing in some cases.

11.   The conclusion section may need to be improved, reiterating the achievements and contributions of the article.
